# Domain Adaptation for Deep Reinforcement Learning in Visually Distinct Games

## Abstract

Many deep reinforcement learning approaches use graphical state representations, this means visually distinct games that share the same underlying structure cannot effectively share knowledge. This paper outlines a new approach for learning underlying game state embeddings irrespective of the visual rendering of the game state. We utilise approaches from multi-task learning and domain adaption in order to place visually distinct game states on a shared embedding manifold. We present our results in the context of deep reinforcement learning agents.

## 1 Introduction

Games have often been used in order to experiment with new approaches with Artificial Intelligence (AI) research. Games provide flexibility to simulate a range of problems such as fully observable vs partially observable, stochastic vs discrete and noisy vs noise free environments. This first started with digital versions of board games being used such as backgammon and chess. More recently video games have begun to provide a plethora of digital environments and tasks for benchmarking AI systems. These new systems use neural networks and are usually trained using the raw pixel values of game frames, meaning the networks have to interpret these pixels into game states that can then be used to learn an optimal policy for play. Due to the fact that they use these raw pixel values they are sensitive to changes in the visuals of the game used, this results in very little knowledge transfer between visually distinct games (Rusu et al., 2016) resulting in the networks learning each game individually without any representation overlap. Games are usually very visually distinct as concepts are often abstract, especially for puzzle games. Early video games were often like this because it is very computationally expensive to create games that accurately imitate the real world, whereas more modern games may take a more abstract representation due to the financial expense.

Learning representations has long been an important area of AI research. It has led to many new approaches to producing representations for many applications, such as word embeddings (Bengio et al., 2003) and style transfer (Gatys et al., 2015). In the case of word embeddings networks can be used to solve various tasks such as predicting the context within which each word appears, the outputs of one of the hidden layers can then be used in order represent that word. This then results in word embeddings that place words that appear in similar contexts within close proximity to each other in the embedding space (Bengio et al., 2003). These resulting embeddings can then be used in order to solve more complex tasks without the system having to train from raw input. This could be seen as a form of knowledge transfer as the knowledge of the meaning of words has been encoded into its embedding and can then be used with new networks working on new tasks without the need to learn this mapping between words and their context again (Turian et al., 2010).

In our work we improve knowledge representation across tasks that have underlying similarities but are represented in a visually distinct way. The architecture we propose in this paper learns representations that are independent of the visual differences of the games. This will result in the strategic elements of the game playing network to share knowledge between visually distinct games.

In order to achieve this we use and extend work that has been done around domain adaption. Domain adaption seeks to produce shared representation between two separate domains a source domain and a target, such as high resolution product photos and images taken with a low resolution webcam (Ganin & Lempitsky, 2015). We present a method for using similar techniques in the domain of reinforcement learning allowing an agent to learn domain independent representations for a group of similar games that are visually distinct.

This paper will first ground our work in the context of both learning representations and reinforcement learning. We will then outline the environment that the networks were trained in along with the desired outcomes. We will finally present the resulting representations and outline future work to extend this approach.

## 2 RELATED WORK

### 2.1 AI AND GAMES

Games provide a good benchmark for AI as they require high level reasoning and planning (Yannakakis & Togelius, 2015; McCoy & Mateas, 2008). This means that they have often been used to signify advances in the state of the art such as with Deep Blue the *chess* playing AI that beat Gary Kasparov (Campbell et al., 2002) and AlphaGo the *go* playing agent that beat Lee Sedol (Silver et al., 2016). Games also provide a nice interface for agents to be able to either look into the internal state of the games, as needed for methods such as Monte carlo tree search (MCTS) (Perez-Liebana et al., 2016), or can provide visual representations of state that can be used by agents (Bellemare et al., 2015). Games also allow agents to process experiences much faster than would be possible in the real world. This means data hungry methods are still able to learn in a relatively short period of time (Karakovskiy & Togelius, 2012).

There has also been a significant amount of research into other areas of game playing, such as competitions aimed at agents passing a form of *Turing Test* in order to rank agents based on their ability to mimic human play (Hingston, 2009). Most of these systems have revolved around using data from human play in order to achieve this goal (Polceanu, 2013; Karpov et al., 2013). There have also been competitions organised in order to assess agents ability to generalise across a variety of games, the General Game Playing Competition (Genesereth et al., 2005) focusing on playing multiple board games, and the General Video Game Playing Competition (Liebana et al., 2016) that assess agents performance across a broad range of video games. Other work has also started that uses modified versions of game engines in order to provide environments to teach AI, these include Project Malmo that uses the popular Minecraft Game in order to provide an experimentation platform for AI agents (Johnson et al., 2016), and OpenArena, a modified version of the ID Tech 3 engine used by Quake 3, in order to train and benchmark AI at a range of tasks including path finding in a Labyrinth and laser tag (Jaderberg et al., 2016).

There has also been some work in using neural evolution to evolve a network to control an agent in an first-person shooter (FPS) environment (Parker & Bryant, 2012). Others have investigated the use of hierarchical approaches in the design of AI agents for FPS games (Van Hoorn et al., 2009) that use networks in a hierarchical fashion to deconstruct the tasks into sub-skills.

### 2.2 REINFORCEMENT LEARNING

Reinforcement learning is an area of machine learning that focuses on the use of rewards in order to train agents how to act within environments. Reinforcement learning provides a structure where the agent is in an environment in which it can take actions, then make observations and receive rewards. These rewards can then be used in order to improve the agent's policy for picking actions in the future, with the aim of maximising the rewards it receives.

Temporal difference (TD) is a technique that has had a large impact in reinforcement learning (Kaelbling et al., 1996). It gives a method for predicting future reward through learning from experience. TD learning is the basis for many other reinforcement approaches (Watkins & Dayan, 1992; Konda & Tsitsiklis, 1999).

Actor-critic is one method the utilises TD. Actor-critic methods split the agent into two separate entities (Williams, 1992). The actor learns a policy which is a vector of probabilities for selecting a specific action. The critic is then a more traditional state-value function that is used to guide the learning of the policy. The critic evaluates the TD error $\delta$ to assess whether the action chosen performed better or worse than expected (Konda & Tsitsiklis, 1999). If the action produced a better reward than expected the probability of choosing that action should be increased. However if the reward was less than expected the probability $p(s_t, a_t)$ of choosing action $a_t$ should be reduced.

This is one of the simpler forms of actor critic methods with other approaches introducing more factors into the calculation such as the entropy of the current policy (Williams & Peng, 1991).

There are two main advantages to actor critic over value function methods, one is that it takes minimal amount of computation in order to select actions. There is no need to calculate the value of $Q$ for every action in the action space before picking the action with the highest value. The other advantage is that they can learn explicitly stochastic policies, meaning it can learn the optimal probabilities for which action to pick. One example of this is that actor critic methods could learn the optimal policy for the game rock, paper, scissors. It would learn that the optimal policy is to pick each action one third of the time, this is the optimal if you are not modelling the decisions of your opponent (Niklasson et al., 2001). Whereas a value function method with a greedy policy would constantly pick the same action depending on the experiences it had during training.

## 2.3 DEEP REINFORCEMENT LEARNING

There have already been a few systems that tackle the problem of learning to play games from the screen buffer only. As will be discussed below all of these systems rely on the use of deep neural networks in order to process the images and learn which action produces the best possible outcome for the agent.

Deep Q networks (DQN) were the first deep reinforcement learning agents to beat human performance in a variety of Atari games (Mnih et al., 2015). This has lead to many advancements including Asynchronous Advantage Actor-Critic networks that are the basis for our experiments.

Asynchronous advantage actor-critic (A3C) is a method for using actor-critic reinforcement learning methods along with deep reinforcement learning, it is also the approach that we chose to use in our experiments. There are a few different intricacies to A3C networks, the first is the fact that the value and policy are calculated by two separate networks. Although in theory they are two different networks, in practice it is possible for the two networks to share lower level layers, that includes all the convolutional layers and any LSTM layers that the model may have. As it is an actor-critic method the policy is learnt online. Meaning the policy is followed during training at all times and there is no need for a explore vs exploit parameter. Instead one of the tricks used by A3C is to subtract the entropy of the policy from the error rate, this tries to force the network towards high entropy until it finds a policy that produces high rewards, this has been shown to find better policies by increasing exploration and preventing the system from converging on a sub optimal policy too early (Williams & Peng, 1991). A3C is also an asynchronous method, with each thread having an instance of the MDP that the agent can learn from. Once a set of experiences have been acquired on a particular thread then the gradients are calculated on the local network in order to then update the global network. Once the global network has been updated a new copy is taken by the local network. This means that A3C can train entirely on CPUs and does not require GPUs to attain acceptable training times (Mnih et al., 2016).

This method negates many of the tricks required by the DQN technique that are there to improve the convergence of those systems. The fact that the local threads take a copy of the global network prevents the need for network freezing, where old copies of the network are used for the error calculation. Having multiple threads all updating the global network with different experiences also means there is no need for experience replay. A3C has been shown to perform better than DQN approaches in a variety of Atari 2600 games (Mnih et al., 2016).

There have also been some improvements made to A3C including the Unsupervised reinforcement and auxiliary learning (UNREAL) networks who use a variety of auxiliary tasks in order to improve feature extraction from the raw input images (Jaderberg et al., 2016).

## 2.4 MULTI-TASK LEARNING

Multi-task learning (MTL) is a method that tries to leverage the similarities between tasks in order to improve training efficiency and accuracy (Caruana, 1998). In the context of deep learning these systems try to share this knowledge by having shared layers of the neural networks used, these are the lower layers of the network that then feed into separate output layers for each task (Caruana, 1998). This results in the lower layers producing representations of the input data that can be used to solve a variety of tasks. Features that are extracted by one task can then be shared among all the

tasks allowing them to leverage the learned features. This form of MTL has many similarities to auxiliary tasks such as the ones used in UNREAL (Jaderberg et al., 2016).

There has been some work with using MTL in the context of deep reinforcement learning. One of these approaches by Liu et al uses DQN as the base reinforcement learning agent and then presents the agent with different but similar tasks. These tasks involve solving a navigation problem within the Malmo environment, this means that the output layers for the different tasks would be dealing with differing action space sizes and different actions to choose between. During training of this network the task would be switched after every training episode and a separate experience replay memory would be stored for each task. This has been shown to produce aligned embeddings across similar tasks with a visually similar environment (Lydia T. Liu & Hofmann, 2016). It has also been shown however that these systems cannot align the embeddings for similar tasks that are visually distinct (Rusu et al., 2016).

## 2.5 DOMAIN ADAPTION

Domain adaption is an approach that tries to deal with differing data sets that have the same underlying structure. These approaches are often applied to image data sets such as the Office dataset that contains images of the same objects split into three data sets, one containing the amazon product images, one containing images taken with a high resolution dSLR camera and finally images of the objects captured using a low resolution webcam. There is also extensive amounts of research in using domain adaption for natural language processing, such as sentiment analysis where product reviews across a variety of domains are used (Glorot et al., 2011). Originally a loss between these tasks was hard coded (Fernando et al., 2013) whereas more recent approaches use adversarial networks in order to produce a loss that can be used to produce a domain adaption embedding space (Ganin & Lempitsky, 2015). Using the Office dataset still shows results based on supervised learning of real world representations. The most diversity in input that the network needs to deal with is differing quality of images and lighting conditions. The objects still have a familiar real world representation that changes very little between data sets.

## 2.6 SUMMARY

The research presented above represents the grounding for our work. In our work we look to improve the representations that deep reinforcement learning agents produce when learning multiple visually distinct games. As shown when visually distinct games are used in the context of multi task learning each game would result in separate embedding spaces Rusu et al. (2016). Domain adaption is an approach that tries to create shared embedding spaces from differing raw datasets. We look to use approaches from domain adaption in order to create unified embedding spaces for visually distinct games in order to facilitate knowledge transfer in the future by taking advantage of a shared embedding manifold.

## 3 APPROACH

For our approach we wish to create a neural network system that can effectively learn to play multiple visually distinct games that share the same underlying structure. Solving this problem opens up the possibility of sharing knowledge between distinct data sources that still describe the underlying structure of similar tasks. This would allow for more general AI systems as well as increasing the amount of data that can be used in order to solve a task, as multiple differing data sources could be used. It may also be possible to use the representations in order to learn new unseen datasets in a more time and data efficient manner than learning from scratch, such as the current systems do.

Our approach retains the advantages of deep reinforcement learning systems by learning to play these games form the raw pixel values and a given reward function only. As our research is concerned with learning these underlying representations we have chosen to use an established deep reinforcement learning approach. We chose A3C given its success in learning a variety of games, but this approach could easily be adapted to other deep reinforcement learning algorithms. In this paper we look at games that have the same underlying structure and game states, however we change the visual representations of these game states so that the network needs to learn the underlying similarities and not the raw pixel value similarities. We will compare this to current domain adaption

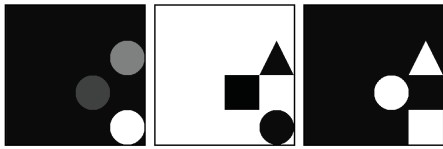

Figure 1: Game Renderers (Left to Right: GreyRenderer, InvertRenderer, ShapeRenderer)

approaches and also the naive approach of simply training the A3C network on all three representations.

Our approach also uses techniques that have shown success in domain adaption (Ganin & Lempitsky, 2015). We use an adversarial network in order to provide an error gradient that tries to force the lower layers of the network into creating embeddings that are agnostic to the dataset that produced them.

## 4   EXPERIMENTAL SETUP

Our work is focused on obtaining a shared embedding manifold between games that have an underlying similarity but which are rendered differently. The environment that we setup in order to experiment involved using the game GridWorld, a simple 2D game where a player character can move in the four cardinal directions and must collect a pickup and then head towards a final goal. We then take the gridworld game and create three renderers that render the game in distinct ways.

The frames shown in Figure 1 all show the same game state. The Shape Renderer renders the player as a circle, the pickup as a triangle and the goal as a square. The Invert renderer has the colors inverted and changes which shape corresponds to the game elements. The Grey renderer uses the greyscale value to discriminate between the game objects. All the games were played on a 4x4 grid with the agent receiving a positive reward of 0.5 for collecting the pickup, and a reward of 1 when reaching the goal having already collected the pickup, the images produced by the renderers were 40x40 pixels in size. The size of the grid was chosen to be 4x4 in order to keep the training time relatively short. Our choice of reward function was based on rewarding the agent only for positive actions towards it's goal, as our research isn't focused on improving the planning of actions or dealing with distant rewards we also added a reward for collecting the pickup in order to speed up convergence to a good policy. Goal and player position are random for every episode of a game. Our goal is to have our system learn to play these games and separate the underlying game state from the visual representation.

## 5   NETWORK DETAILS

For all our experiments we compare our approach to two other approaches, we keep as much of the networks the same in order to test our enhancements. We will outline our approach and then highlight the differences between our two benchmarks.

### 5.1   DOMAIN ADATATION FOR DEEP REINFORCMENT LEARNING (DA-DRL)

Our approach is shown in Figure 2. The architecture is a standard A3C implementation, but instead of training on one game it is fed multiple games with the upper layers being shared. This is to try and get the upper layers to embed knowledge of how to play the games independent of how they are rendered. The input images are all 8bit greyscale and are 40x40 pixels in size. The convolutional layers for all the games are the same in size but do not share any weights, the first layer has 16 8x8 kernels that have a convolution step size of 4, the second layer has 32 4x4 kernels that have a step size of 2. The upper hidden layer has 16 neurons and the action output is a probability distribution of size 4, one for each direction the agent can move. There is also a value estimation output that is used during training in order to guide the agents policy.

The error gradients for the A3C elements of the network are calculated using

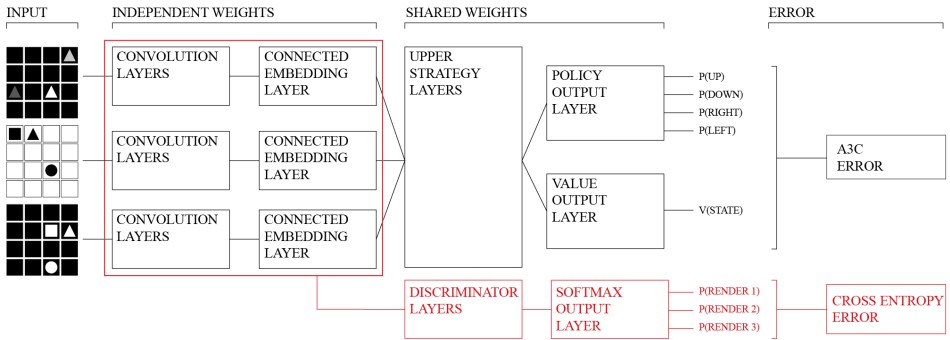

Figure 2: Network Architecture

$$\nabla_{\theta'} \log \pi(a_t|s_t;\theta')(r_t - V(s_t;\theta_v)) + \beta \nabla_{\theta'} H(\pi(s_t;\theta')) \tag{1}$$

where $\pi(a_t|s_t;\theta')$ is the output of the policy network for action $a_t$ in state $s_t$, $r_t$ represents the actual reward received by the agent, $V(s_t;\theta_v)$ is the estimated reward from the value network, $H$ is the entropy and $\beta$ is the strength of the entropy regularization term. The action taken is factored by the amount of error in the prediction of the value of the state. This has the effect of increasing the probability of actions that it underestimates the value of the states they produce, and reduces the probability of actions that it overestimates the value of the states.

The architecture also has another network that needs to be optimized, in this case an adversary that receives a game state embedding as an input and must predict which renderer produced that embedding. This network consists of an input layer of size 16, a hidden layer of size 16 and then a softmax output layer of size 3 with one output per renderer during training. The cross entropy loss is then minimized during training in order for the network to improve at classifying which renderer produced which embedding. Whilst this is happening the convolutional layers are also updated in order to try and maximise the loss from the discriminator network. This has the effect of the convolutional layers being forced to remove renderer specific data from the embedding that the discriminator sees and only include data that can be extracted from all of the renderers.

Simply adding the discriminator would however not fully force the network to have to remove all render specific information as the convolutional layers can begin to deliver unseen embeddings in order to fool the discriminator into not knowing which renderer produced the embedding. This can be fixed by placing a small error to the embeddings during training that penalises distance from the global distributions of all embeddings across all three renderers. The equation for this penalty is given by

$$\mathcal{L}_E = \frac{1}{n} \sum_{i=0}^{i=n} \bar{E} - e_i \tag{2}$$

where $n$ is the number of embeddings produced in this training batch, $\bar{E}$ is the rolling average of the global set of embeddings and $e$ represents an individual embedding in the training batch.

The loss for the convolutional layers now contains a variety of errors that need to be optimized. The loss calculation for these layers are calculated using

$$\mathcal{L}_{CONV}(\theta) = \mathcal{L}_{A3C} + \lambda_E \mathcal{L}_E - \lambda_A \mathcal{L}_A \tag{3}$$

where $\mathcal{L}_{A3C}$ is the A3C loss as calculated by Equation 1, $\lambda_E \mathcal{L}_E$ being the discounted embedding loss as outlined in Equation 3 and $\lambda_A \mathcal{L}_A$ being the discounted adversarial networks loss in the case of our network this is the cross entropy loss. The adversarial networks loss is is subtracted from

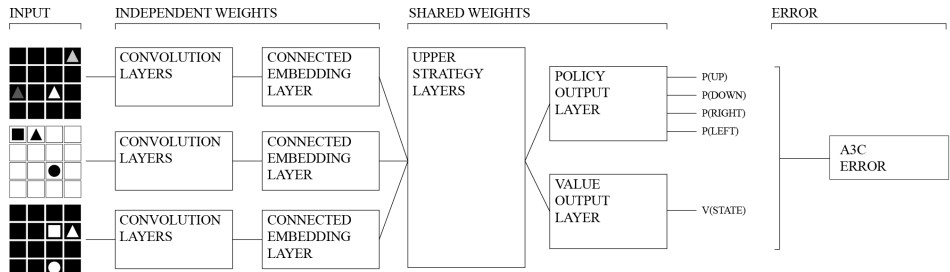

Figure 3: This Figure shows the architecture for our baseline

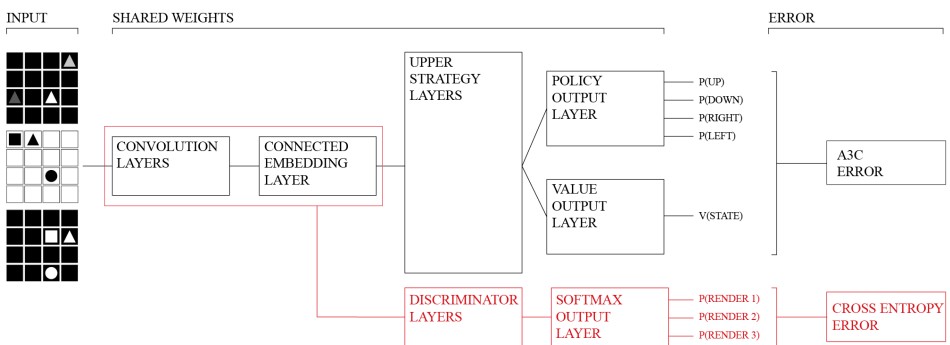

Figure 4: This Figure shows the network with completly shared parameters betweeen games.

the convolutional layers loss so that the convolutional layers are trying to maximise the adversarial networks error.

We expect that our system would be able to deal well with generalisability due to the fact that the bases of the reinforcement learning is A3C that has been shown to generalise well when dealing with a wide variety of games. We also use a relatively small $\lambda_A$ when calculating the gradients for the convolutional layers. This should allow for games that do not share underlying similarities to diverge within the embedding space, this however has not yet been tested.

## 5.2 BASELINE

The first variation we will compare to is an identical network to our approach but without the adversarial network, this will be used to demonstrate whether the embeddings between the separate tasks naturally align into a shared manifold. All of the other parameters are identical to our approach above.

## 5.3 SHARED CONVOLUTION

The final system we test against is show in Figure 4. This network shares the lower convolutional layers between all three games such as the networks by Ganin & Lempitsky (2015)

## 6 RESULTS

As shown in Figure 5 the network with fully shared parameters was unable to reach the same performance as our network with separate convolutional layers per renderer. With our network managing to get very close to optimal performance in around 3 million examples. The fully shared network however was unable to achieve these results only getting to 10 average excess moves per game. The baseline system was able to get to our performance faster but without managing to extract similarities between game states as shown in Figure 6 where each visually distinct game is fully disconnected in the embedding space. These results are from a single training run, but we have found that these

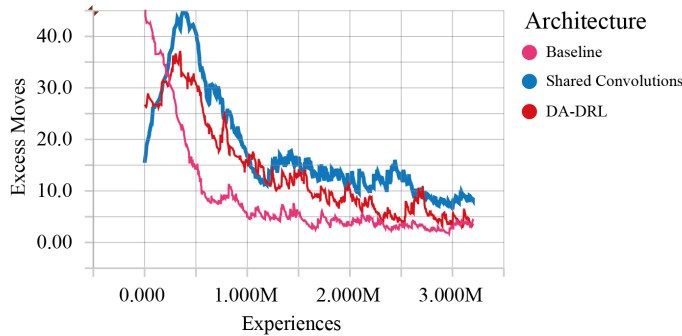

Figure 5: Average game play performance during training

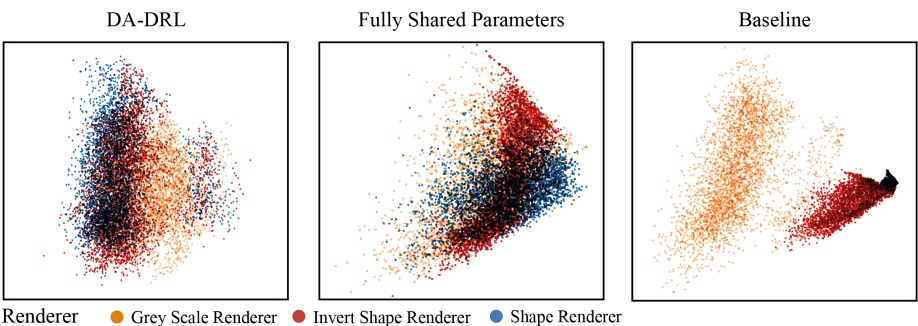

Figure 6: Embedding results colored by renderer used for each data point, plotted using Principle Component Analysis (PCA)

results are consistent across multiple runs, with only the initial performance of the networks being affected due to random weight initialisation. Interestingly the performance of both the DA-DRL network and the shared convolution network decreases in the early stage of training, we believe this is due to an untrained adversarial network having a bigger influence at the start of training.

Looking at Figure 6 it is obvious that there is some overlap between the renderers however it is possible to see that our system better aligns the separate renderers as they follow the same overall structure, whereas in the fully shared parameter version the renderers embeddings have differing shapes and cross over each other at differing angles. Also looking at Figures 7 and 8 it can be seen that there isn't as much structure to the embeddings in terms of encoding the best action to take or if there is a pickup present in the game state.

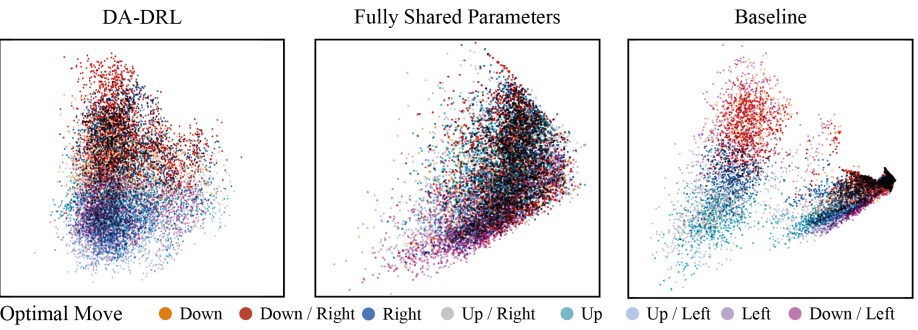

Figure 7: Embedding results colored by best possible move for each data point, plotted using PCA

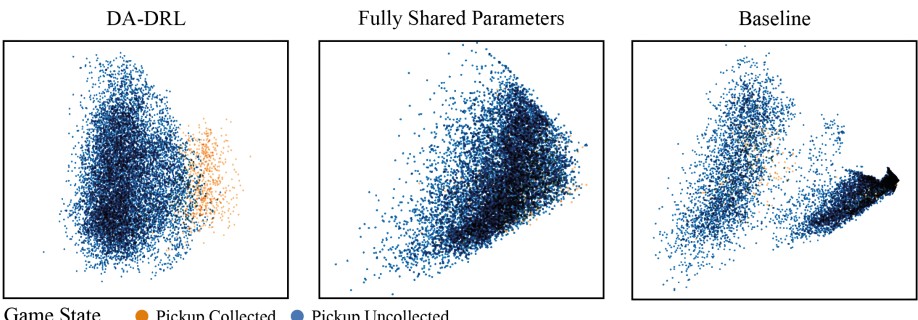

Figure 8: Embedding results colored by the presence of the pickup in game state, plotted using PCA

Whereas the network with separate convolutional layers per renderer was able to successfully align its embeddings as shown in Figure 6 and also encode information about both the optimal move to make and the current state of the game, in this case whether the game state has a pickup present or not. This is not a property that we tried to impose during training but rather an emergent property of the embeddings that the network produced.

## 7    CONCLUSION

As our results show it is possible for our system to separate game states from raw pixel values despite the renderings of the game being so distinct. It also effectively separates out states that require a different task to be completed, with the game states that contain pickups being in a distinct but related space to the states that require the player to now head towards the goal. This shows that the learned embeddings are capturing a lot about how the game is played irrespective of how the frame is rendered. We show that this is possible in the context of deep reinforcement learning for domain adaptions to be successfully achieved using adversarial networks. Our network also manages to deal with more visually diverse inputs than was possible with a network that fully shares its parameters. As the results have shown the single shared convolutional layers did not have the capacity to deal with the differences between the visual representations of the game. We have shown that it is possible to use adversarial networks along with separate convolutional layers in order to produce shared embedding spaces for visually distinct inputs.

## 8    FUTURE WORK

In the future we plan to use the knowledge we have extracted from this approach to allow the network to effectively utilise these embeddings in order to improve learning on a new, but visually distinct, game. We believe the knowledge that has been extracted would aid in the data efficiency of learning a new task that is visually distinct but shares similarities with the original task. We would also like to increase the complexity of the underlying games as well as reducing the similarity between the tasks. One identified experiment would be to use 3 games from the Arcade Learning Environment (ALE) that provides an environment for reinforcement learning from screen capture on a variety of Atari 2600 games. We have identified 4 similar yet visually distinct games that could be used in our experiments. Zaxxon, Assault, Beam Rider and Space Invaders all require the player to move some form of entity at the bottom of the screen and shoot upwards in order to destroy entities coming from the top of the screen. They are visually distinct and in some cases also diverge slightly in how then are played, with Space Invaders being the only game with defences at the bottom of the screen. . By achieving this knowledge transfer we would provide a flexible method for the transfer of knowledge between these deep neural network systems, irrespective of their architectures.

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
