# OpenReview forum: "Domain Adaptation for Deep Reinforcement Learning in Visually Distinct Games"
_ICLR.cc/2018/Conference — Reject_

### Official Review · AnonReviewer1 · 2017-11-10
**This paper contains interesting ideas, but it is not ready for publication.**

**Rating:** 3
**Confidence:** 3

**Review:**

In this paper, the authors propose a new approach for learning underlying structure of visually distinct games.

The proposed approach combines convolutional layers for processing input images, Asynchronous Advantage Actor Critic for deep reinforcement learning task and adversarial approach to force the embedding representation to be independent of the visual representation of games.

The network architecture is suitably described and seems reasonable to learn simultaneously similar games, which are visually distinct. However, the authors do not explain how this architecture can be used to do the domain adaptation.
Indeed, if some games have been learnt by the proposed algorithm, the authors do not precise what modules have to be retrained to learn a new game. This is a critical issue, because the experiments show that there is no gain in terms of performance to learn a shared embedding manifold (see DA-DRL versus baseline in figure 5).
If there is a gain to learn a shared embedding manifold, which is plausible, this gain should be evaluated between a baseline, that learns separately the games, and an algorithm, that learns incrementally the games.
Moreover, in the experimental setting, the games are not similar but simply the same.

My opinion is that this paper is not ready for publication. The interesting issues are referred to future works.

---

### Official Review · AnonReviewer2 · 2017-11-27
**Interesting idea maybe? Very poor experimental section.**

**Rating:** 2
**Confidence:** 4

**Review:**

This paper introduces a method to learn a policy on visually different but otherwise identical games. While the idea would be interesting in general, unfortunately the experiment section is very much toy example so that it is hard to know the applicability of the proposed approach to any more reasonable scenario. Any sort of remotely convincing experiment is left to 'future work'.

The experimental setup is 4x4 grid world with different basic shape or grey level rendering. I am quite convinced that any somewhat correctly setup vanilla deep RL algorithm would solve these sort of tasks/ ensemble of tasks almost instantly out of the box.

Figure 5: Looks to me like the baseline is actually doing much better than the proposed methods?

Figure 6: Looking at those 2D PCAs, I am not sure any of those method really abstracts the rendering away. Anyway, it would be good to have a quantified metric on this, which is not just eyeballing PCA scatter plots.

---

### Official Review · AnonReviewer3 · 2017-11-28

**Rating:** 4
**Confidence:** 5

**Review:**

- This paper discusses an agent architecture which uses a shared representation to train multiple tasks with different sprite level visual statistics. The key idea is that the agent learns a shared representations for tasks with different visual statistics

- A lot of important references  touching on very similar ideas are missing. For e.g. "Unsupervised Pixel-level Domain Adaptation with Generative Adversarial Networks", "Using Simulation and Domain Adaptation to Improve Efficiency of Deep Robotic Grasping", "Schema Networks: Zero-shot Transfer with a Generative Causal Model of Intuitive Physics".

- This paper has a lot of orthogonal details. For instance sec 2.1 reviews the history of games and AI, which is besides the key point and does not provide any literary context.

- Only single runs for the results are shown in plots. How statistically valid are the results?

- In the last section authors mention the intent to do future work on atari and other env. Given that this general idea has been discussed in the literature several times, it seems imperative to at least scale up the experiments before the paper is ready for publication

---

### Decision · Program_Chairs · 2018-01-29
**ICLR 2018 Conference Acceptance Decision**

**Decision:**

Reject

**Comment:**

The reviewers have found that while the task of visual domain adaptation is meaningful to explore and improve, the proposed method is not sufficiently well-motivated, explained or empirically tested.